# Adolescent mental health research in Tanzania: a study protocol for a priority setting exercise and the development of an interinstitutional capacity strengthening programme

Angela Obasi [iD] ,[1] Maaike Seekles [iD] ,[1] Judith Boshe [iD] ,[2] Dorothy Dow [iD] ,[3,4] Blandina Mmbaga,[4,5] Fileuka Ngakongwa,[6] Elialilia Okello,[7] Jenny Renju [iD] ,[8,9] Elizabeth Shayo,[10] Gema Simbee,[11] Jim Todd,[12,13] Ndekya Oriyo[14]

AO and MS are joint first authors.

For numbered affiliations see end of article.

**Correspondence to**
Maaike Seekles;
maaike.seekles@lstmed.ac.uk

## ABSTRACT

**Introduction** Poor adolescent mental health is a barrier to achieving several sustainable development goals in Tanzania, where adolescent mental health infrastructure is weak. This is compounded by a lack of community and policy maker awareness or understanding of its burden, causes and solutions. Research addressing these knowledge gaps is urgently needed. However, capacity for adolescent mental health research in Tanzania remains limited. The existence of a National Institute for Medical Research (NIMR), with a nationwide mandate for research conduct and oversight, presents an opportunity to catalyse activity in this neglected area. Rigorous research priority setting, which includes key stakeholders, can promote efficient use of limited resources and improve both quality and uptake of research by ensuring that it meets the needs of target populations and policy makers. We present a protocol for such a research priority setting study and how it informs the design of an interinstitutional adolescent mental health research capacity strengthening strategy in Tanzania.

**Methods and analysis** From May 2021, this 6 month mixed-methods study will adapt and merge the James Lind Alliance approach and validated capacity strengthening methodologies to identify priorities for research and research capacity strengthening in adolescent mental health in Tanzania. Specifically, it will use online questionnaires, face-to-face interviews, focus groups, scoping reviews and a consensus meeting to consult expert and adolescent stakeholders. Key evidence-informed priorities will be collaboratively ranked and documented and an integrated strategy to address capacity gaps will be designed to align with the nationwide infrastructure and overall strategy of NIMR.

**Ethics and dissemination** National and institutional review board approvals were sought and granted from the National Health Research Ethics Committee of the NIMR Medical Research Coordinating Committee (Tanzania) and the Liverpool School of Tropical Medicine (United Kingdom). Results will be disseminated through a national workshop involving all stakeholders, through ongoing collaborations and published commentaries, reviews, policy briefs, webinars and social media.

---

**Strengths and limitations of this study**

► Systematic inclusion of all stakeholders, including policy makers
► Nationwide approach and dissemination strategy using NIMR infrastructure
► Transparent and integrated approach to research priority setting and capacity strengthening
► Inclusion of adolescent stakeholders aged 18+ limits applicability to younger adolescents at this stage but will be the focus of future activity

## INTRODUCTION

Mental health disorders account for a high burden of disease among young people worldwide. They are responsible for three of the top six causes of disability-adjusted life years lost among 10–24 years old[1] and affect up to 20% of all adolescents.[2] Suicide is the third leading cause of death in this age group. Adolescents who experience depressive disorders and other mental health problems are more likely to engage in risky behaviours, such as unprotected sexual activity and substance misuse. They are also more likely to have low educational attainment, to be unemployed and to live in poverty.[3] Much of adult mental ill-health begins in adolescence, with an estimated 75% of all lifetime mental disorders occurring by the age of 25.[4] The burden of mental ill-health among adolescents and its potential impact on adult outcomes makes adolescence a critical time for mental health promotion, early identification of mental health disorders and rapid access to effective adolescent mental health (AMH) care.[2 5 6] This is especially important in Africa, which will be home to around 35% of the world's adolescents by 2050.[7]

**BMJ**

In Tanzania, where around half of the population is under 20 years of age, there are several important barriers to effective prevention and care for AMH disorders.[8] First, research on the prevalence, causes and solutions for several AMH issues is lacking.[9 10] The few available studies, mainly on depression and alcohol use disorders, indicate that AMH issues are common in Tanzania.[8 11] Second, AMH disorders are often not recognised as illnesses and stigmatisation of mental ill-health presents a further barrier to care.[12 13] Third, AMH is a low policy priority.[14] In 2017, there were 0.06 psychiatrists and 0.01 psychologists per 100 000 population; this is well below average for the Africa Region.[15] Currently, Tanzania has only one psychiatrist specialised in child and AMH disorders (Dr Gema Simbee, 2021). Further, primary and community-based healthcare providers, such as community health workers, general nurses and community health officers, often have received very limited training, if any, in the delivery of mental healthcare to adolescents.[16 17]

Health research is an essential tool for addressing health and development challenges and for informing health policy. Careful prioritisation of research to maximise its likely impact and the efficient use of resources is especially important where resources are scarce.[18] Of equal importance is the capacity to conduct high-quality research to address identified priorities. Limited research capacity in low and middle income countries is a recognised barrier to improved health and health systems.[19–21] Research capacity in Tanzania faces both significant challenges and opportunities.[22] In particular, the existence of a national, publicly funded Institute for Medical Research provides the opportunity for systematic, nationwide scale-up of both research and capacity strengthening. With eight research centres across Tanzania, the National Institute for Medical Research (NIMR) is the largest research institute in Tanzania. It is mandated by the government to conduct and drive health research nationally, develop national health research priorities and oversee research implementation.

Against this background, the present study will first conduct rigorous research priority setting (RPS) to identify current AMH research gaps and priorities in Tanzania. In line with best practice, perspectives of policy makers, implementers and adolescent advocates will be thoroughly canvassed to ensure priorities meet the needs of all stakeholders. This will include a mapping of AMH service implementation within the government and civil society sector, including professional associations and internationally funded programmes. Second, we will concurrently map and identify gaps in existing AMH research capacity. This will inform the development of a strategy to ensure sufficient research capacity to address identified research gaps and priorities, including optimal use of the opportunities provided by NIMR, other research institutions and Civil Society Organisations (CSOs) within the country.

## Study aims and objectives

The study aims are twofold: first, to identify gaps and priorities in AMH research in Tanzania which will inform the codevelopment of an evidence-based, stakeholder-driven research agenda; and second to identify gaps in capacity needed to deliver this research agenda, which will inform the development of a national research capacity strengthening (RCS) programme to address these gaps.

The specific research objectives are as follows:
1. Identify and bring together leading researchers, implementers and agencies, policy and decision-makers, and other AMH stakeholders in Tanzania.
2. Summarise and catalogue available evidence on the AMH burden and on the implementation and evaluation of AMH interventions in Tanzania.
3. Identify research priorities and develop a future research agenda for AMH.
4. Map existing AMH research capacity and identify gaps in capacity needed to deliver the research agenda.
5. Design an RCS programme to address AMH research priorities.

## METHODS AND ANALYSIS
### Rationale for selection of methodologies

There is considerable variation in approaches, tools and methods used in health RPS.[23] Recent WHO RPS guidance[24] identifies four phases from planning to evaluation (figure 1) and discusses a range of tools and approaches which can be used in each phase. The guidance recommends use of comprehensive approaches to improve the quality and transparency of the exercise.[24] One widely recommended approach, which has been used previously in Africa,[25] is a methodology designed by the James Lind Alliance (JLA).[26]

Well-described and freely available, it consists of four steps which are aligned to the planning and implementation phases of the WHO cycle as follows (see figure 2):
1. Initiation: Establish a group of stakeholders.
2. Gathering uncertainties (consultation): Via a questionnaire, stakeholders are asked to present their list of possible research priorities.
3. Verifying uncertainties (collation): The research team theme the possible priorities. A literature search is done to check whether there is any existing research evidence on the possible priorities.
4. Prioritisation: During a priority setting workshop, involving discussions in breakout groups and the stakeholder group as a whole, a top 10 of research priorities will be agreed on.

The JLA methodology ensures a transparent and rigorous approach to RPS. However, it focuses specifically on identifying priorities for research on the effects of treatment interventions. Since our priority setting exercise has a broader scope, not all aspects of the JLA process are applicable, or possible. The current research will instead adapt and build on the JLA methodology and

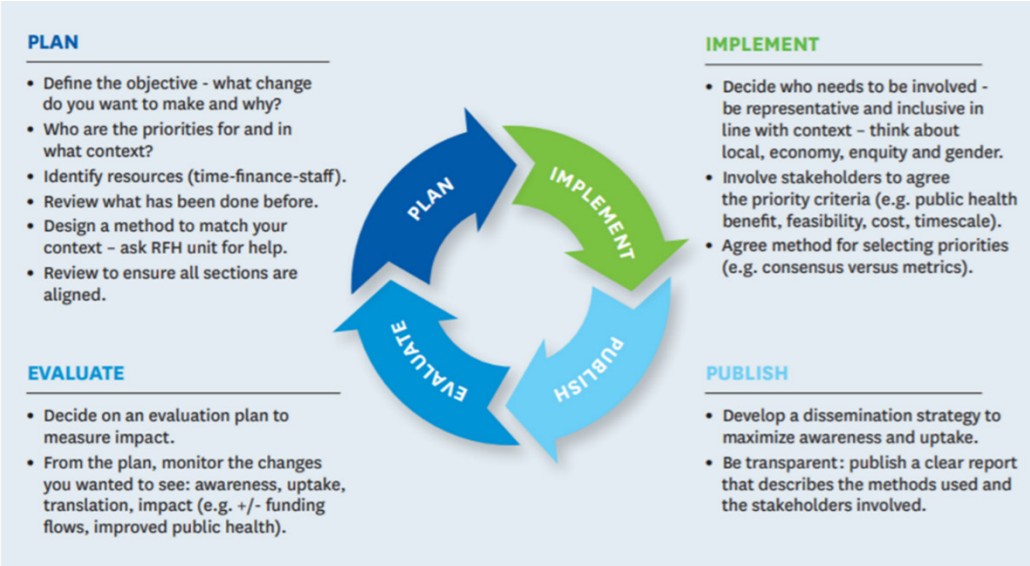

**Figure 1** The RPS process. Reproduced with permission from WHO,[24] licence: CC BY-NC-SA 3.0 IGO. RFH, research for health; RPS, research priority setting.

combine it with activities related to RCS as the second component of this study.

Our RCS will use methods recommended by Bates *et al* which have been validated in a number of research institutions in Africa[27] and consist of the following steps,[27] the first four of which fall within the scope of this study:

1. Define the goal of the capacity strengthening programme.
2. Describe the required capacity needed to achieve the goal.
3. Determine the existing capacity and identify any gaps compared with the required capacity.

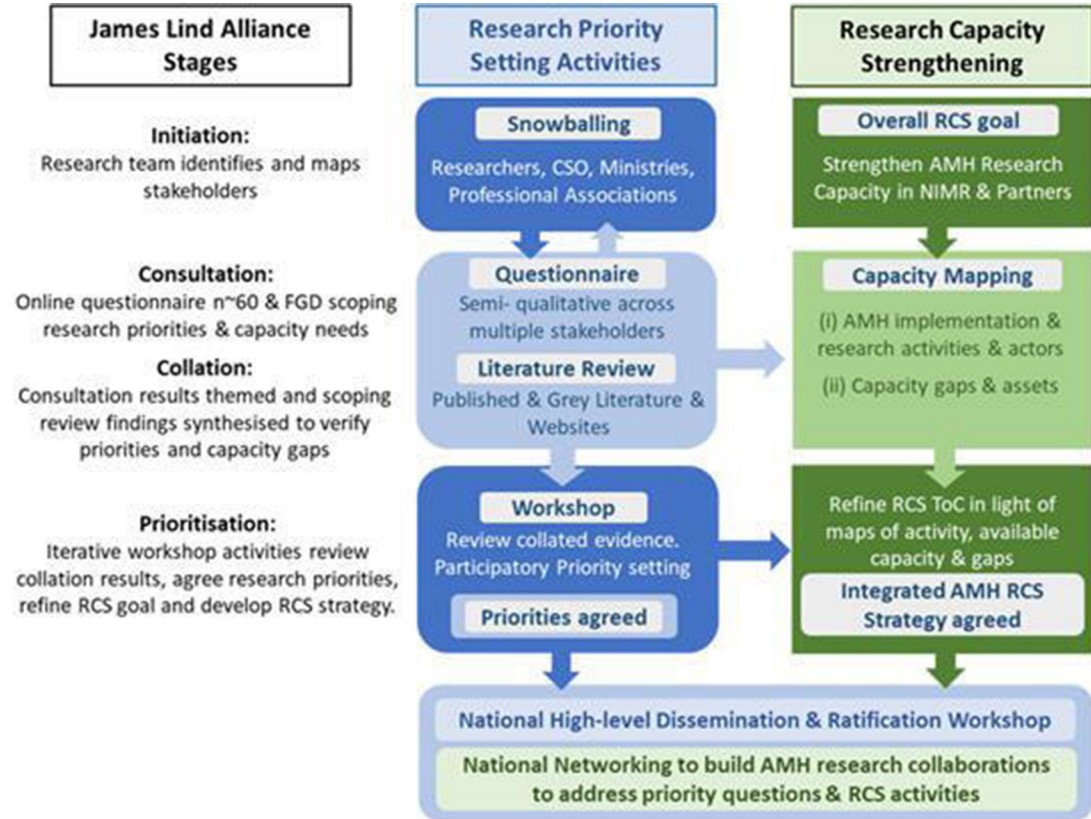

**Figure 2** Project flow diagram. AMH, adolescent mental health; CSO, Civil Society Organisation; FGD, focus group discussion; NIMR, National Institute for Medical Research; RCS, research capacity strengthening; RPS, research priority setting; ToC, Theory of change.

4. Devise and implement an action plan to fill the gaps, setting out activities, cost implications and indicators of progress.
5. Learn through doing; adapt the plan and indicators regularly.

This protocol details the application of the RPS and the modification of the RCS methods and processes that will be implemented to achieve the study objectives.

## Detailed study design

This mixed methods study will be conducted over 6 months, from May to December 2021, in five overlapping phases. It will include online questionnaires, scoping reviews of the literature, individual interviews, focus group discussions and workshops. RPS and RCS activities will run concurrently where appropriate (see figure 2). This protocol has been prepared using the Standard Protocol Items: Recommendations for Interventional Trials Guidelines (SPIRIT, see online supplemental file 1), where applicable.

## Participants

All stakeholders in the field of AMH in Tanzania will be eligible to participate if they are 18 years of age or older. This includes active researchers, implementers, agencies, youth advocacy groups and any relevant policy makers. For logistical reasons including prevailing COVID-19 restrictions and the lack of resources needed to create safe spaces for younger adolescents currently receiving treatment, adolescents aged below 18 years could not be included. However, this study will inform the design of a future priority setting activity that will focus specifically on younger adolescent services users and their carers.

## Patient and public involvement

This preparatory consultation study is constrained by limited resources, the challenges of the ongoing COVID-19 pandemic and the complexities related to access to vulnerable AMH service users. The study largely applies the methods as described in the JLA RPS methodology but departs from the approach in terms of study population. A JLA RPS exercise typically involves patients, carers and clinicians.[26] Our study is limited to professional stakeholders and youth advocates aged 18 or over. As explained above, engaging younger patients is outside the scope of this study, but will be a main subject of investigation in future follow-on studies when COVID-19 constraints have been relaxed and funds are available to ensure safety and support. This study will, however, gain the input from young people and service users in the community by including representatives of youth advocacy groups, whose expert views are of pivotal importance.

## Processes

### Phase 1: identification and recruitment of stakeholders (Spring 2021)

The initiation phase aims to identify and bring together stakeholders in the field of AMH. As a starting point, the research team, which includes researchers from NIMR, Kilimanjaro Christian Medical University College, Research Institute and Medical Centre and Muhimbili University of Health and Allied Sciences, will list any active researchers, implementers and agencies, and youth advocacy groups that they are aware of in Tanzania. In addition, relevant policy makers working in the field of AMH will be identified. The identification of stakeholders will be an iterative process throughout this study. As a project deliverable, a heat map will be created of identified AMH researchers and implementers across the country.

### Phase 2: initial consultation (Summer 2021)

The second phase aims to 'gather uncertainties'[26] and gains an initial insight into stakeholder views on priorities and capacity gaps in AMH research.

#### Questionnaire

The research team will distribute a link to an online questionnaire (using Survey Monkey) to all identified stakeholders. It is expected that around 60 participants will be reached. A sample size calculation was not appropriate, since all identified stakeholders will be invited to participate. The questionnaire will include a consent statement. This will be followed by a mix of open-ended and closed questions aimed at capturing the range and coverage of current AMH activities, in addition to priorities for research. The questions will be adapted slightly by type of stakeholder, but will generally address current involvement in the field of AMH service provision or research; perceived gaps in AMH research skills/capacity and gaps in provision of AMH care. In addition, participants will be asked to list three AMH research priorities (see online supplemental file 1 for the questionnaire sent to researchers).

To ensure that all active stakeholders are reached through a process of snowballing, respondents will be asked to nominate other relevant stakeholders. Respondents will also be invited to signal their willingness to participate in the remainder of the RPS and RCS process. The data collection will take place over a period of 1 month commencing May 2021. In case of non-response, to maximise participation, identified stakeholders will receive two reminders via email. Because of the high likelihood that stakeholders who do not work in a research environment will have poor internet connectivity, non-researchers will receive reminders via phone calls and/or a text message. The study will also be publicised through research and implementer networks (see figure 3).

Once the survey is closed, members of the research team will come together to review, interpret and sort participant priorities. During discussions, responses will be turned into a list of indicative questions for research, or 'evidence uncertainties'.[26] These will be entered into a data management spreadsheet. Questions will be framed according to the Population, Intervention, Control, Outcome format where possible. Out-of-scope submissions will be removed.

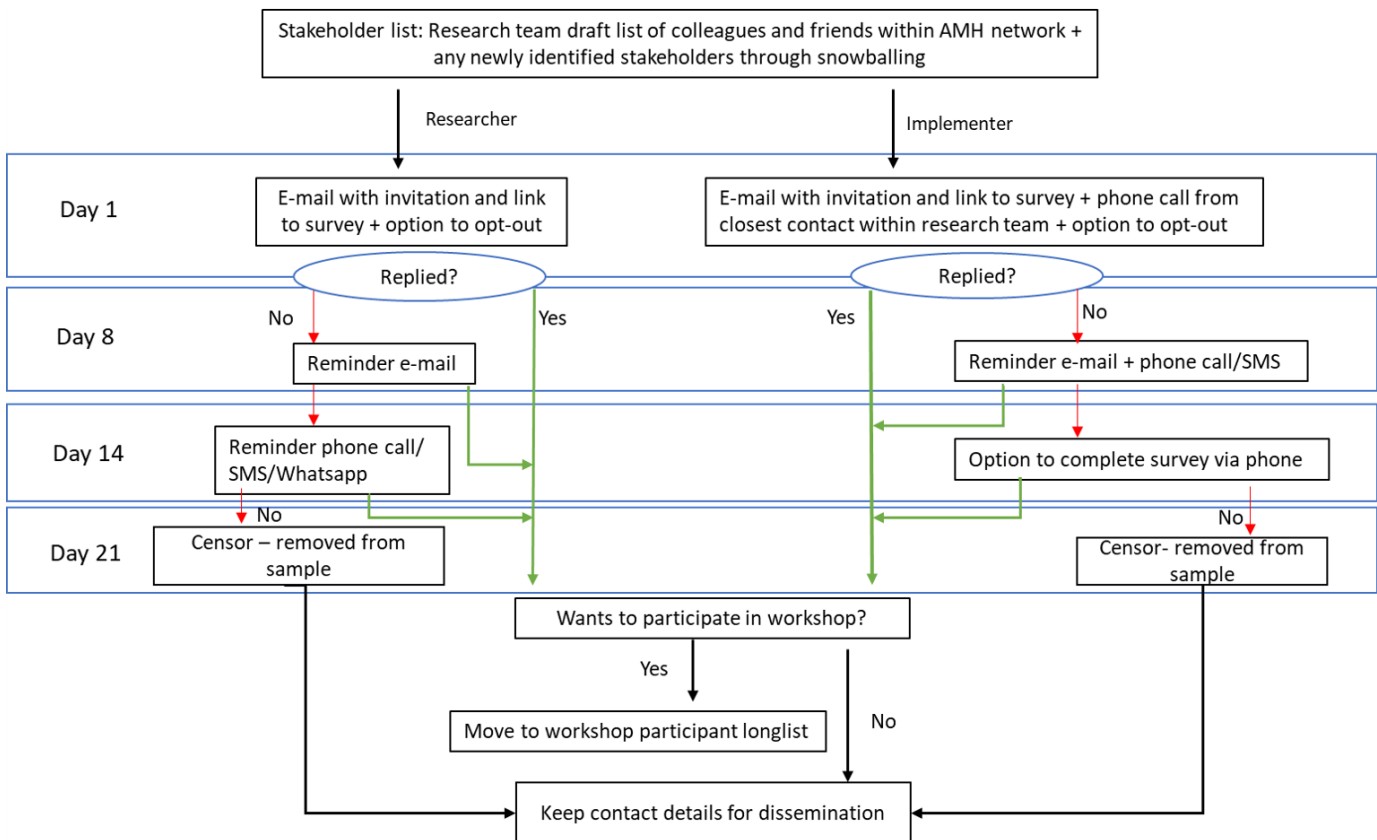

**Figure 3** Survey and workshop recruitment flow chart. AMH, adolescent mental health.

## Phase 3: data processing and verifying uncertainties

The JLA and Bates *et al* recommend the use of a literature review in an early stage of RPS and RCS exercises.[26 27] For the RPS, this is to verify that each indicative question identified in the survey is a true uncertainty. For each indicative question, a search in the Cochrane Database of Systematic Reviews and Liverpool School of Tropical Medicine's (LSTM's) library resources will be done to identify systematic and scoping reviews that might address the uncertainty. Any recommendations for further research identified in the reviews will be noted and may be included as priorities in the spreadsheet. Only up-to-date reviews, published in or after 2017, that focus on adolescent research in sub-Saharan Africa will be explored. For each research question, a question verification form[26] will be completed to show how the research team concluded that the question is broadly unanswered.

## Phase 4: Scoping reviews (Spring/Summer 2021)

Initial searches have already identified a gap in relevant AMH literature reviews in sub-Saharan Africa. In preparation of the priority setting exercise, we will undertake two scoping reviews, on AMH promotion/prevention and treatment interventions. A scoping review does not aim to produce a critically appraised and synthesised answer to a precise question related to a certain treatment or practice. It rather aims to provide an overview or map of the available evidence.[28] It can report on the types of evidence and the way research has been conducted,

which aligns with the interests of the research team. The scoping reviews will follow the 5-step framework of Arksey and O'Malley.[29] They will include scientific and grey literature about studies with adolescents, undertaken in sub-Saharan Africa from 2000 onwards. Searches will be done in English and Kiswahili.

## Phase 5: prioritisation and RCS programme development (Autumn 2021)

The final phase aims to synthesise the information collected through the above activities and bring stakeholders together to design a research agenda and develop an RCS programme.

### Workshop

After analysis of the questionnaire and initial scoping reviews, a workshop will synthesise research priorities and refine the goal for the RCS programme. Participants will be selected from the larger stakeholder group based on their availability and whether they registered an interest to take part. It is estimated that around 25 stakeholders, including national and subnational representatives of the Ministry of Health and local government, will be able to participate. This falls within the range recommended by the JLA of 12–30 participants per workshop.[26] Should more than 30 stakeholders want to participate, the research team will purposively select participants to ensure an appropriate representation of all stakeholder groups. During the 2 day workshop, group discussions will

take place, focusing on current research and perceived gaps in research, research capacity and care for AMH.

Per JLA guidance, an adapted Nominal Group Technique will be used as the method for RPS. Participants will first form smaller discussion groups of maximum eight participants, during which each participant gives their view about which priorities are most and least important, followed by shared ranking or voting. The ranked orders for each item from each group are totalled, and the priority with the lowest, that is, most favoured, total ranking is selected as the top priority. The ranking is then discussed and reviewed in the large group, to arrive at a final list of 10 research priorities.[26] The final priorities will inform the development of a research agenda on AMH, with key research questions outlined in collaboration with partners.

The remainder of the workshop will focus on RCS. During group discussions, a process of joint problem-solving with stakeholders will identify strengths, prioritise critical capacity gaps, and transform what evidence indicated was optimal into what is feasible and practical. The gaps, reasons for the gaps, discrepancies, and resolutions, and potentially sustainable solutions for filling capacity gaps are mapped onto a matrix.[27] The team, together with participants, will then use this matrix to underpin the design of a theory of change and an RCS programme. The RCS strategy will align activities for strengthened capacity, for example, mentoring, PhD in specific areas, skills development to potential studies and study partnerships, with identified research priorities. Intrainstitutional and interinstitutional partnerships and additional resources needed to deliver these activities will be agreed.

During workshop days, a small number of key informant interviews (n~5–10) and focus group discussions (eg, with representatives of national professional organisations and youth advocates) will take place, to verify identified priorities in light of local knowledge about context, epidemiology and ongoing activities. Focus groups and interviews will be conducted in English or Kiswahili. They will be recorded and transcribed verbatim.

## ETHICS AND DISSEMINATION
National and institutional review board approvals were sought and granted from the National Health Research Ethics Committee of the NIMR Medical Research Coordinating Committee in Tanzania (Reference number: NIMR/HQ/R.8a/Vol. IX/3670) and the LSTM in the United Kingdom (Reference number: 21–002).

Participation in this study is voluntary. All stakeholders will receive information about the study, to allow them to make an informed decision about their participation. Survey participants are asked to confirm approval with a consent statement. At the workshop, participants will be asked to sign an informed consent form. They have the right to withdraw at any time by informing the Principal Investigator. However, participants are advised that once their data is anonymised, it cannot be removed. Due to

the nature of the focus group discussions and workshops, confidentiality cannot be guaranteed, but participants will be reminded to not share any sensitive data that can bring harm on themselves or others.

A COVID-19 Risk Assessment, detailing appropriate precautions to limit the spread of COVID-19, has been completed. This will be updated throughout the study duration, to ensure that research activities comply with latest regulations.

A national dissemination meeting will be organised in Tanzania once the strategy and priority setting is finalised and written up, during which the results of the RPS exercise and RCS assessment will be presented. The developed national AMH research agenda will be launched as a platform for collaborative, interinstitutional research programmes which will embed RCS activities. The findings will be further disseminated through papers, commentaries, reviews and policy briefs.

## Data management and analysis
This study will comply with LSTM and NIMR data protection policies.[30 31] All electronic data and personal information of participants will be stored on password protected computers and secure drives, only accessed by designated members of the research team as appropriate. Group discussions and interviews will be recorded on an encrypted device; recordings will be transcribed verbatim as soon as possible after data collection, after which recordings will be deleted. Data will be stored for a maximum of 10 years, to allow verification of data from external sources if necessary, or longer if used for further research.

Data from online questionnaires will be extracted from Survey Monkey and simple descriptive statistical analysis will be undertaken, using Excel software to provide summaries of the sample and responses. In addition, answers to open questions will be coded and analysed through team discussions using content analysis to arrive at an initial list of gaps in research and capacities.

Workshop data and data collected during group discussions and interviews will be thematically analysed, using NVivo software, to offer a more in-depth account of arguments for or against research priorities, and explanations and possible solutions for capacity constraints and research gaps.

## DISCUSSION
To summarise, this study will use a mixed-methods approach to consult experts, policy-makers and advocates and identify priorities and capacity gaps for AMH research in Tanzania. It gains its strengths from a multidisciplinary partnership between, and the involvement from inception of, key stakeholders from a number of research institutions across Tanzania; the use of a systematic, transparent approach to RPS; an evidence-based approach to RCS which is aligned to priority research activities; and a strong commitment to dissemination.

As discussed earlier, this study departs from the JLA method in terms of study population. It also departs from the approach in terms of scope of investigation. The JLA approach usually focuses on uncertainties around treatment for a specific illness.[26] The aim of this study is to identify priorities on a national level, across the spectrum of AMH research. This could include uncertainties around prevalence, prevention and treatment of varying illnesses, in addition to wider systemic challenges. By engaging key policy makers and the leadership and structure of the NIMR, it is hoped this study will raise the profile of AMH research within national research policy. The interinstitutional RCS partnerships generated through this process are also aligned to other best practice strategies for equitable partnership such as the WHO ESSENCE guidance.[32] In particular, it is hoped that the opportunities provided for AMH researchers, representatives from CSOs, development partners, youth advocates and senior government to meet and network during this study, will significantly strengthen AMH research structure and legacy.

**Author affiliations**
[1]Department of International Public Health, Liverpool School of Tropical Medicine, Liverpool, UK
[2]Psychiatry and Mental Health, Kilimanjaro Christian Medical Centre, Moshi, United Republic of Tanzania
[3]Division of Infectious Diseases, Department of Pediatrics, Duke University Medical Center, Durham, North Carolina, USA
[4]Kilimanjaro Christian Medical Centre, Moshi, United Republic of Tanzania
[5]Kilimanjaro Clinicial Research Institute, Moshi, United Republic of Tanzania
[6]Department of Psychiatry and Mental Health, Muhimbuli National Hospital, Dar es Salaam, United Republic of Tanzania
[7]Mwanza Intervention Trials Unit, National Institute for Medical Research Mwanza Research Centre, Mwanza, United Republic of Tanzania
[8]The London School of Hygiene & Tropical Medicine, London, UK
[9]Epidemiology and Biostatistics, Kilimanjaro Christian Medical University College, Moshi, United Republic of Tanzania
[10]Department of Policy Analysis and Advocacy, National Institute for Medical Research, Dar es Salaam, United Republic of Tanzania
[11]Mirembe National Psychiatric Hospital, Dodoma, United Republic of Tanzania
[12]National Institute for Medical Research Mwanza Research Centre, Mwanza, United Republic of Tanzania
[13]Department of Population Health, London School of Health and Tropical Medicine, London, UK
[14]National Insititute for Medical Research, Dar es Salaam, United Republic of Tanzania

**Contributors** AO and NO are co-principal investigators who jointly secured funding for, initiated and led conception of the study. They share overall responsibility for the research. All authors (AO, NO, MS, JB, DD, BM, FN, EO, JR, ES, GS and JT) attended project meetings, and contributed to the design of the study and data collection materials. As joint first authors, AO and MS led the drafting of the paper. All other authors (NO, JB, DD, BM, FN, EO, JR, ES, GS and JT) provided critical input for revisions and approved the final version. With the exception of AO, NO and MS, authors are listed alphabetically.

**Funding** This study is funded by a networking grant from the Global Challenges Research Fund (UKRI, Ref: GCRFNGR6_1454). The funder has no involvement in the design; collection, management, analysis, and interpretation of data; writing of the report, and the decision to submit the report for publication, nor do they have ultimate authority over any of these activities.

**Competing interests** None declared.

**Patient consent for publication** Not applicable.

**Provenance and peer review** Not commissioned; externally peer reviewed.

**ORCID iDs**
Angela Obasi http://orcid.org/0000-0001-6801-8889
Maaike Seekles http://orcid.org/0000-0002-7000-3624
Judith Boshe http://orcid.org/0000-0002-1684-7295
Dorothy Dow http://orcid.org/0000-0002-9056-1025
Jenny Renju http://orcid.org/0000-0001-5650-1902

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
