## [Reviewer comments · BMJ Open]

ARTICLE DETAILS

TITLE (PROVISIONAL)	Adolescent mental health research in Tanzania - a study protocol for a priority setting exercise and the development of an inter-institutional capacity strengthening programme.
AUTHORS	Obasi, Angela; Seekles, Maaïke; Boshe, Judith; Dow, Dorothy; Mmbaga, Blandina; Ngakongwa, Fileuka; Okello, Eliaililia; Renju, Jenny; Shayo, Elizabeth; Simbee, Gema; Todd, Jim; Oriyo, Ndekya

VERSION 1 – REVIEW

REVIEWER	Sharma, Tarang The Nordic Cochrane Centre-Rigshospitalet, 7811
REVIEW RETURNED	28-Sep-2021

GENERAL COMMENTS	General: As a protocol this should have all details of the methods expected clearly described but there are no clear specifics or details of any of the sections. This clearly seems to be a very important piece of research and therefore the planning/ protocol should be well thought through and designed. To not have that described would be a shame. 1. Major: The methods section noted the JLA method along with the four steps from the WHO priority setting guide. However, some of the details for the individual components is missing. For example, the survey is open to several participants and using a further snowball approach was initiated in May 2021, so a sample could be shared as an appendix, as it is September, and could be part of the protocol submission. There is very little information about the content of the survey, focus groups and interviews, some more detail of what it is the researchers exactly are hoping to get would be helpful and how they came up with the questions or content would be good to ascertain. Normally a sample of the interview/ focus group guide, could also be shared as an appendix. This would have already been done I suspect for ethical approval, so it is a shame to not have that for the publication. 2. Major: Additionally, it is interesting that the scoping reviews are being conducted later, and not to inform the interview guides or focus group questions (as it is normally done for priority setting) but rather all considered together at the end. Some explanation or rationale of this choice by the authors would be helpful. 3. Major: Again, the details of the scoping review methods are completely missing – there is no PICO or equivalent structure informing how it is expected to be done, what the review question and parameters would be. Is it searching reviews only or primary literature, or is it going to have a timeframe or not. Nothing is clear. 4. Major: Again, it is not very clear how the data derived from the questionnaire, interviews, focus groups and the scoping review will
--

	all be then distilled into the research priorities list to be used in the JLA process. Having attended one JLA process for their prostate cancer priority setting many years back, the data is derived from the database of research uncertainties – so the input to the process is evidenced- informed and of sufficient quality. The data entering the database is quality assured and checked. Having done several more priority setting cases since in other institutions, the process of bringing the data into research questions is important and needs to be transparent. Will there be a any mechanism of quality check and distillation of thoughts into research topics – please describe that within the protocol. 5. Minor: The inclusion of all stakeholder groups is noted a few times in the manuscript and is clearly what is needed, having a full inclusive approach. However, they talk about starting with the research team, and then some implementers, policy makers and youth advocacy groups. There are again no specifics here – some stakeholder mapping of the relevant policy makers, youth organizations working in this field of adolescent mental health that they would like to target would be good to know.
--	---

REVIEWER	Town, Rosa University College London
REVIEW RETURNED	04-Oct-2021

GENERAL COMMENTS	This is a well-written and engaging study protocol. There are a few grammatical details which I have outlined below. Overall, really well done and I look forward to reading the results of this research. Line 225: remove comma before 'by' Line 285: comma needed after 'reviews' Line 295: 8 should be written 'eight' Line 349: remove comma after 'approach'
---

VERSION 1 – AUTHOR RESPONSE

Reviewer 1: Dr. Tarang Sharma, The Nordic Cochrane Centre-Rigshospitalet

Comments to the Author:

General: As a protocol this should have all details of the methods expected clearly described but there are no clear specifics or details of any of the sections. This clearly seems to be a very important piece of research and therefore the planning/ protocol should be well thought through and designed. To not have that described would be a shame.

We would like to thank the reviewer for taking the time to thoroughly assess our manuscript. We have addressed your constructive feedback as detailed below, and feel that this has significantly improved the quality of our manuscript.

1. Major: The methods section noted the JLA method along with the four steps from the WHO priority setting guide. However, some of the details for the individual components is missing. For example, the survey is open to several participants and using a further snowball approach was initiated in May 2021, so a sample could be shared as an appendix, as it is September, and could be part of the protocol submission. There is very little information about the content of the survey, focus groups and interviews, some more detail of what it is the researchers exactly are hoping to get would be helpful and how they came up with the questions or content would be good to ascertain. Normally a sample

of the interview/ focus group guide, could also be shared as an appendix. This would have already been done I suspect for ethical approval, so it is a shame to not have that for the publication.

A sample questionnaire has now been added to the appendix. The content of the survey is detailed in sentences 249-252. Further text has been added to clarify what we hope to get out of the survey. This section now reads:

The questionnaire (see appendix A for the researcher questionnaire) will include a consent statement. This is followed by a mix of open-ended and closed questions aimed at capturing the range and coverage of current AMH activities, in addition to priorities for research. The questions will be adapted slightly by type of stakeholder, but will generally address current involvement in the field of AMH service provision or research; perceived gaps in AMH research skills/capacity; and gaps in provision of AMH care. In addition, participants are asked to list three AMH research priorities. Once the survey is closed, members of the research team will come together to review, interpret, and sort participant priorities using a mix of quantitative and thematic analysis. During discussions, responses will be turned into a list of indicative questions for research, or 'evidence uncertainties'.

The focus group discussion guide will be informed by the results of the survey, it is therefore unfortunately not possible to provide this as an appendix at protocol stage.

2. Major: Additionally, it is interesting that the scoping reviews are being conducted later, and not to inform the interview guides or focus group questions (as it is normally done for priority setting) but rather all considered together at the end. Some explanation or rationale of this choice by the authors would be helpful.

Apologies, the interviews and focus groups will indeed take place after the literature searches and scoping reviews, during or shortly after the workshop days. This section had been placed under the wrong heading in error and has now been moved under phase 5 to reflect this.

3. Major: Again, the details of the scoping review methods are completely missing – there is no PICO or equivalent structure informing how it is expected to be done, what the review question and parameters would be. Is it searching reviews only or primary literature, or is it going to have a timeframe or not. Nothing is clear.

Thank you. We have adapted this section so that it more closely follows the JLA methodology of completing database searches for literature reviews and subsequently completing the data management file. The exact parameters are difficult to define, since they depend on the survey responses, but we have defined a timeframe and population and context. Sentences 270-288 now read:

“During discussions, responses will be turned into a list of indicative questions for research, or 'evidence uncertainties'. These will be entered into a data management spreadsheet. Questions will be framed according to the Population, Intervention, Control, Outcome (PICO) format where possible. Out-of-scope submissions will be removed.

Phase 3: Data processing and verifying uncertainties

The JLA and Bates et al. recommend the use of a literature review in an early stage of RPS and RCS exercises.[26, 27] For the RPS, this is to verify that each indicative question identified in the survey is a true uncertainty. For each indicative question, a search in the Cochrane Database of Systematic Reviews and LSTM's library resources will be done to identify systematic and scoping reviews that might address the uncertainty. Any recommendations for further research identified in the reviews will be noted and may be included as priorities in the spreadsheet. Only up-to-date reviews, published in or after 2017, which focus on adolescent research in sub-Saharan Africa will be explored. For each

research question, a Question Verification Form (REF) will be completed to show how the research team came to the conclusion that the question is broadly unanswered.”

In addition, as per our original draft, we will complete scoping reviews which we feel are most important in the current context. The subjects of these reviews have been determined since our original submission, so this has been updated. Details of the scoping review methodology remain the same; they will follow the Arksey and O’Malley framework, which provides a detailed 5-step process. We have now added further details about the type of literature included, timeframe, language of search, and country of study for these reviews. This section now reads:

“Initial searches have already identified a gap in relevant AMH literature reviews in sub-Saharan Africa. In preparation of the priority setting exercise, we will undertake two scoping reviews, on AMH promotion/prevention and treatment interventions. A scoping review does not aim to produce a critically appraised and synthesised answer to a precise question related to a certain treatment or practice. It rather aims to provide an overview or map of the available evidence.[28] It can report on the types of evidence and the way research has been conducted, which aligns with the interests of the research team. The scoping reviews will follow the 5-step framework of Arksey and O’Malley.[29] They will include scientific and grey literature about studies undertaken in sub-Saharan Africa from 2000 onwards. Searches will be done in English and Kiswahili.”

4. Major: Again, it is not very clear how the data derived from the questionnaire, interviews, focus groups and the scoping review will all be then distilled into the research priorities list to be used in the JLA process. Having attended one JLA process for their prostate cancer priority setting many years back, the data is derived from the database of research uncertainties – so the input to the process is evidenced- informed and of sufficient quality. The data entering the database is quality assured and checked. Having done several more priority setting cases since in other institutions, the process of bringing the data into research questions is important and needs to be transparent. Will there be a any mechanism of quality check and distillation of thoughts into research topics – please describe that within the protocol.

The additions made in relation to the comment 3 above also address this comment. Sentence 252 to 257 and 270 to 280 (copied above) now more clearly explain the process of how participant priorities will be turned into indicative questions, which will be verified as true uncertainties using literature searches. For increased transparency, a Question Verification Form will be completed to show how the research team came to the conclusion that the research question is unanswered. Then, policymakers, researchers, implementers and youth advocates will share their views during consensus meeting and a focus group, which will be used to confirm or refute the priorities, based on their knowledge about the context, epidemiology and local knowledge about what is already ongoing.

Sentence 334-338 now read:

“During workshop days, a small number of key informant interviews (n~5-10) and focus group discussions (e.g., with representatives of national professional organisations and youth advocates) will take place, to verify identified priorities in light of local knowledge about context, epidemiology and ongoing activities.”

5. Minor: The inclusion of all stakeholder groups is noted a few times in the manuscript and is clearly what is needed, having a full inclusive approach. However, they talk about starting with the research team, and then some implementers, policy makers and youth advocacy groups. There are again no specifics here – some stakeholder mapping of the relevant policy makers, youth organizations working in this field of adolescent mental health that they would like to target would be good to know.

A stakeholder mapping is one of the deliverables of this study, linked to objective 1. The process is described in the paragraph under phase 1: identification and recruitment of stakeholders. As described, the mapping is an iterative process and therefore is not complete at this protocol stage. A sentence has been added to explain more clearly that the work phase 1 is done to create a map of stakeholders as a study output.

Sentences 239 – 241 now read:

“In addition, relevant policy makers working in the field of AMH will be identified. The identification of stakeholders an iterative process throughout this study. As a project deliverable, a map will be created of identified AMH researchers and implementers across the country”

Reviewer: 2

Dr. Rosa Town, University College London

Comments to the Author:

This is a well-written and engaging study protocol. There are a few grammatical details which I have outlined below. Overall, really well done and I look forward to reading the results of this research.

We would like to thank the reviewer for their feedback.

Line 225: remove comma before 'by'

Comma removed

Line 285: comma needed after 'reviews'

Comma added

Line 295: 8 should be written 'eight'

This is now written as 'eight'

Line 349: remove comma after 'approach'

Comma removed.

VERSION 2 – REVIEW

REVIEWER	Sharma, Tarang The Nordic Cochrane Centre-Rigshospitalet, 7811
REVIEW RETURNED	22-Nov-2021

GENERAL COMMENTS	The authors have added enough detail within the manuscript but mainly by the addition of the supplementary file, the protocol details the methods planned now so my concerns have been addressed.
---